# Diagnostic Techniques for Electrical Discharge Plasma Used in PVD Coating Processes

Sergey Grigoriev [1], Sergej Dosko [2], Alexey Vereschaka [2,*], Vsevolod Zelenkov [2] and Catherine Sotova [1]

1   VTO Department, Moscow State University of Technology STANKIN, Vadkovsky per.1, 127055 Moscow, Russia
2   Institute of Design and Technological Informatics of the Russian Academy of Sciences (IDTI RAS), 127055 Moscow, Russia
*   Correspondence: dr.a.veres@yandex.ru

**Abstract:** This article discusses the possibilities of two methods for monitoring Physical Vapor Deposition (PVD) process parameters: multi-grid probe, which makes it possible, in particular, to determine the energy distribution of ions of one- or two-component plasma and spectrum analyzer of the glow discharge plasma electromagnetic radiation signal based on the Prony–Fourier multichannel inductive spectral analysis sensor. The energy distribution curves of argon ions in the low-voltage operation mode of ion sources with closed electron current have been analyzed. With a decline in the discharge current, the average ion energy decreases, and the source efficiency (the ratio of the average ion energy W to the discharge voltage U) remains approximately at the same level of $W/U \approx 0.68, \ldots, 0.71$ in the operating voltage range of the source. The spectrum analyzer system can obtain not only the spectra at the output of the sensor, but also the deconvolution of the spectrum of the electromagnetic radiation signal of the glow discharge plasma. The scheme of a spectrum analyzer is considered, which can be used both for monitoring and for controlling the processing process, including in automated PVD installations.

**Keywords:** electrical discharge plasma; analyzing methods; Physical Vapor Deposition (PVD); energy distribution





## 1. Introduction

### 1.1. Method of Physical Vapor Deposition and Issues of Process Diagnostics

The processes of ion-plasma treatment and the deposition of coatings through the Physical Vapor Deposition (PVD) technology are actively used in various industries with the aim to increase surface hardness, resistance to wear and corrosion, and control of tribological parameters and optical properties of the products [1–5].

The PVD methods are an effective way to improve the working efficiency and prolong service life of metal-cutting tools and various parts of tool, chemical, machine-building, automotive, aviation, and medical industries. The methods of ion-plasma treatment of surfaces have a number of advantages over other coating deposition methods, including Chemical Vapor Deposition (CVD), spraying with electricity, thermal spraying, as well as electrolytic and galvanic methods [4,5]. In particular, the PVD method provides perfect adhesion to a wide range of various materials, both conductive and dielectric ones. The advantages of the PVD method also include the possibility of depositing high-quality coatings at considerably low temperatures (compared to the CVD methods), the possibility of using a wide range of coating compositions and structures, and the possibility of effectively controlling the parameters of a coating during its deposition [4,5].

When any method of surface modification and coating deposition is used, the key task is to ensure the quality and reproducibility of the results of the technology process carried out in accordance with the selected method. The PVD methods are based on the impact of high-energy flows of gas or metal plasma on the products. This exposure

changes the structure and properties of the coating surface and its energy state, the degree of which depends on the type and mode of the treatment. The result of such exposure depends on many parameters that are difficult to control, including the chemical and phase composition and the structural state of the substrate material, which change during the exposure to the plasma flow [6–8]. The listed parameters directly depend on the energy characteristics of the plasma flow. When developing new technological devices for the deposition of PVD coatings and ion-plasma surface treatment, as well as for the control of the technology ion-plasma processes, it is necessary to use the methods of diagnostics and dynamic measurement of various physical properties of the surface and the parameters of plasma flows. Some of these methods are considered below.

*1.2. Key Methods of Operational Control of the Deposition Process Parameters*

The development of the PVD technologies and the modification of surfaces of solid bodies requires the study methods providing for the quick control of the state of the process and the results obtained at various stages of technology cycles.

Although the process of ion-plasma treatment of surfaces by the PVD method applies such standard control techniques as measurement of the arc current and bias voltage, the composition and pressure of the reaction gas, and the turntable rotation speed [9,10]. Due to the complexity of the ion-plasma treatment process and the variety of influencing factors, the mentioned control parameters are often not enough to ensure unambiguous reproducibility of the process results. The results of ion-plasma treatment are significantly affected by such factors as the spatial arrangement of samples, sizes and weight of technological equipment, the cathode wear degree, the condition of inner surface of the chamber, and a number of other parameters, the direct control of which is difficult or impossible [6,11–17]. Thus, there is a need for dynamic control of the plasma parameters and the state of the modified surface.

With regard to the operational control over the parameters of the ion-plasma treatment and the state of modified surfaces, the following considerably effective methods may be considered [18–30]:

- Langmuir probe methods [19–22];
- application of a charged particle energy analyzer (multigrid probe) [23–25];
- and use of a broadband spectrum analyzer based on a multichannel inductive sensor and Prony–Fourier (PF) spectral analysis.

Various technological ion-plasma processes of surface treatment and coating deposition in vacuum are determined mainly by the parameters of the plasma acting on the surface and the temperature of the very workpieces. Therefore, due to a change in the parameters of the environment, one technological cycle can cover various processes: an increase of the ion energy makes it possible to conduct ion cleaning, and a decrease of the ion energy ensures the deposition of coatings. The energy of ions bombarding the surface of the substrate is one of the most important parameters to be measured and controlled during the development of cleaning and etching technologies for various materials, as well as the PVD technologies [31–36]. The plasma density (concentration) and the temperature of electrons and ions determine the equilibrium temperature of the treated workpieces and, consequently, the diffusion processes and the processes of crystal structure growth on the surface of the treated workpieces.

The main parameters of the low temperature plasma generated by various electric discharges used in the technological processes of treating the surfaces of the workpieces in vacuum (glow discharge, magnetron discharge, vacuum arc discharge with a sacrificial cathode, etc.) include plasma density, electron temperature, ion energy, and ion current density on the surface of the workpieces [37,38]. The non-equilibrium nature and high-power density make plasma a considerably complex object to control and study.

There is a large number of diagnostic methods for determining various parameters of both high temperature and low temperature plasma. Of them, the most widely used are electromagnetic, spectral, microwave, and probe methods [39–43].

The spectral methods give accurate information for optically thin plasma, while for optically dense plasma, the information content of the method is low. Some alternative diagnostic methods are discussed in [44–46].

One of the most common methods used in scientific research and in the industrial practice are microwave methods [38–40]. Scattering of external high-frequency electromagnetic radiation on free electrons leads to a change in the frequency of scattering wave, the measurements of which determine the distribution function of particles according to their speeds, density, temperature, and directed velocity. Probe methods of diagnostics are the most common and easy to implement, allowing to determine the main parameters of ion-plasma flows necessary for the control of the PVD technological processes.

Physical processes in technological installations are often oscillatory in nature, so the universal model proposed by Bulgakov [47] can be used to describe them, and the Prony method [48,49] is the most acceptable for model identification in terms of physical adequacy and accuracy of approximation.

Estimating the parameters of an exponential signal is an important task, since the response of a linear system to an impulse action is the sum of just such signals. Thus, by evaluating the parameters of the signals at the output of the system, it is possible to solve the problem of identifying the system and its state. The use of the Fourier transform for this purpose does not always give acceptable results. This is due to the fact that the Fourier transform is intended to estimate the signal spectrum, not the frequency, and, moreover, in the classical version it is not statistically stable [50].

For non-stationary oscillations, spectral analysis is required, with which you can:

1. perform spectral estimation of segments of time series without side effects in time windows of limited duration;
2. use a non-stationary time series model (for example, increasing or decreasing in the time window);
3. determine own frequency spectrum of a segment of the time series;
4. and determine damping at natural frequencies.

Considering the above requirements, it seems promising to use the Prony method to determine the time-dependent damping spectra of non-stationary oscillations. When identifying the similarity of segments of a time series, there are not a number of restrictions inherent in the Fourier transform of time series [50].

The Langmuir probe measurements [19–22,41,51–59] are widely used low temperature plasma diagnostic methods. This method makes it possible to determine the main parameters of the low temperature plasma in the local region, including electron temperature, electron density, and ion current density, and, in some cases, to approximately estimate the ion temperature. Probe measurements are reliable, provided that the lengths of the particle path exceed the dimensions of the probe. Interpretation of data in the presence of considerably large magnetic fields and collisions is complicated. In accordance with the method Langmuir probe measurements, a small electrode—a probe—is placed in the plasma (to reduce disturbances caused in the plasma, the probe should be small) and its current-voltage curve is taken (the probe current value depending on its voltage), which is called the plasma probe characteristics. Probes can be in the form of a flat disk, a cylinder, or a sphere. A typical diameter is from $10^{-3}$ to $10^{-1}$ cm for a cylindrical probe and from $10^{-2}$ to $10^{-1}$ cm for a spherical probe. To take the plasma probe characteristics, it is important that the electrons have the energy distribution close to Maxwellian distribution, which is provided by the plasma of vacuum stationary discharges used in technological processes. The scope of the method for gas pressure covers the range from $10^{-2}$ to $10^{2}$ Pa and from $10^{7}$ to $10^{5}$ cm$^3$—for the concentration of charged particles.

This article discusses the possibilities of two methods for monitoring PVD process parameters:

- multi-grid probe, which makes it possible, in particular, to determine the energy distribution of ions of one- or two-component plasma generated by a vacuum arc evaporator;

- and spectrum analyzer of the glow discharge plasma electromagnetic radiation signal based on the Prony–Fourier multichannel inductive spectral analysis sensor.

## 2. Materials and Methods

### 2.1. The Langmuir Probe Measurements

To determine the parameters of the ionic component, a charged particle energy analyzer (multigrid probe) was used [60–65]. Figure 1 exhibits the method for measuring the energy of charged particles using a retarding potential energy analyzer [66].

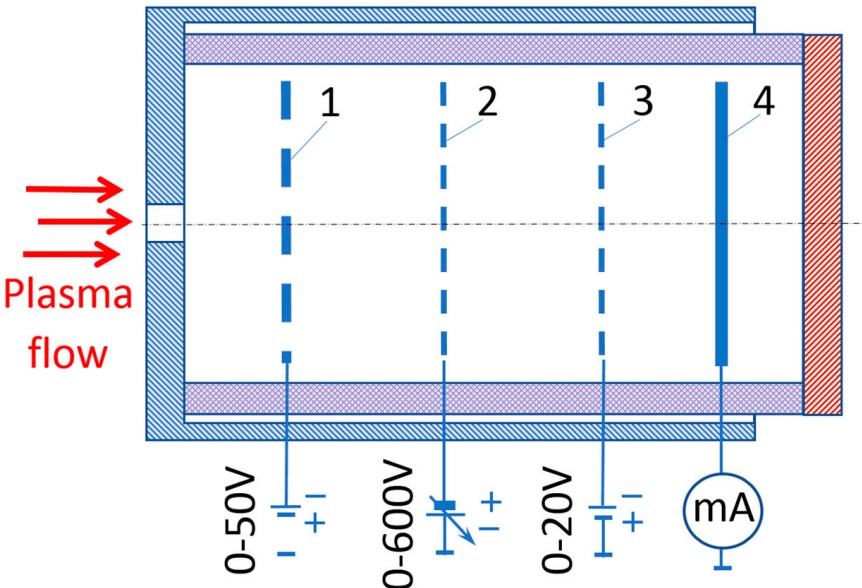

**Figure 1.** Schematic diagram of multigrid analyzer probe [59]. 1—electron-intercepting grid, 2—analyzing (retarding) grid, 3—antidynatron grid, 4—collector.

In the course of measurements, the plasma enters the analyzer through the hole, and the increasing retarding potential, which decelerates the ions, is applied to the analyzing grid 2, while the electrons are intercepted by the grid 1. The ions with energy higher than the retarding potential barrier of the grid 2 form the current of the collector 4. Based on the measurement results, a retarding parameter—a curve of the collector current depending on the retarding potential—is being plotted. To determine the energy distribution function of particles (differential energy spectrum), it is necessary to differentiate the resulting retarding curve. To improve the accuracy of measurement, an antidynatron grid is used in the probe to suppress the secondary emission of electrons that occurs under the action of ion bombardment of the collector surface.

The disadvantages of the described method include:

- high sensitivity of the probe to its contamination;
- and when working with metal plasma, probes quickly fail, or their readings change due to the dust deposited on the input grid; the application of probes is complicated in the presence of considerably large magnetic fields in the plasma.

At high plasma densities, when the free paths of electrons and ions are short, the probe method is practically not applicable.

From the point of view of vacuum technologies for modifying surfaces of solid bodies, the advantage of the probe methods of analysis is in the possibility of integrating the probes, due to their small sizes, into the technological volumes and the possibility of carrying out operational control and adjusting the technological process.

A multigrid probe was used to determine the energy distributions of ions in the two-component plasma, generated by a vacuum arc evaporator with two evaporating electrodes

similar to those described by Zelenkov et al. [67]. The device consists of an evaporable cathode, an evaporable anode, and a passive (non-evaporable) anode (see Figure 2).

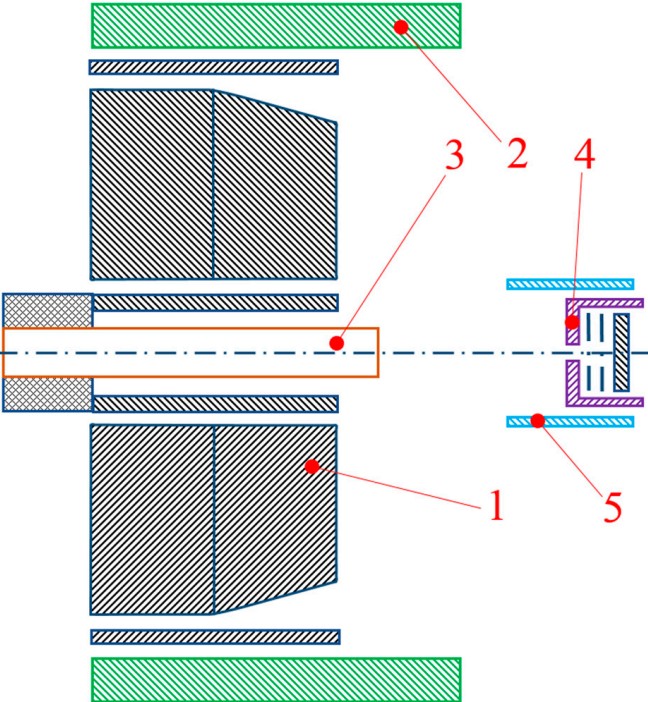

**Figure 2.** Scheme of multigrid probe [67]. 1—cathode, 2—passive anode, 3—evaporable anode, 4—probe (energy analyzer), 5—screen.

*2.2. Broadband Spectrum Analyzer*

An effective method for analyzing the plasma electromagnetic radiation signal (in particular, the glow discharge plasma) is to use a broadband spectrum analyzer based on a multichannel inductive sensor, analog-to-digital converter (ADC), and Prony–Fourier (PF) spectral analysis. The mentioned spectrum analyzer may be used both for monitoring and for controlling the ion-plasma process of surface treatment of solid bodies, including in automated technological environments, with high reproducibility of results. Changes in the shape, amplitudes, and frequencies of the recorded electromagnetic pulses during the treatment process are closely related to the pulsations of the current in the discharge during the technological process of treatment. In particular, such method may be used to determine the moment when the PVD unit reaches the operating mode, to control the stages of surface thermal activation and coating deposition, as well as to promptly respond to deviations from the specified parameters of the technological process.

The wide use in recent decades of the resonance methods for the study of substances in gaseous, liquid, and solid states is justified by their universality [68–73]. The concept of resonance means an increase in the response of an oscillating system to a periodic external action when its frequency approaches one of the natural frequencies of the system. All oscillating systems are able to resonate and may have a very different nature. In the substances, such systems may be electrons, electron shells of atoms, magnetic and electric moments of atoms and molecules, impurity centers in crystals and individual crystals, as well as their groups. However, in all cases, the general picture of the resonance is preserved, that is, near the resonance, the amplitude of the oscillations and the energy, transferred to the oscillatory system from the outside, increase. This increase stops when the energy losses compensate for its gain.

Resonance methods can be attributed to the most sensitive and accurate methods for studying the substances (and, consequently, affecting them) [68–70]. The resonance methods allow obtaining various types of information about the chemical composition,

structure, symmetry, and internal interactions between the structural units of a substance. A substance, depending on its internal structure, has its own unique set of natural oscillation frequencies (frequency or energy spectrum). Natural frequencies $f_k$ can be in a wide range from $10^2$ to $10^{22}$ Hz. Such sets of frequencies is a kind of visiting card of the substance. Electromagnetic radiation is a common and effective type of periodic external action. The frequencies of electromagnetic waves are in the ranges of $10^2 \ldots 10^8$ Hz (radio waves), $10^9, \ldots, 10^{11}$ Hz (radio microwaves), $10^{13}, \ldots, 10^{14}$ Hz (infrared light), $10^{15}$ Hz (visible light), $10^{15}, \ldots, 10^{16}$ Hz (ultraviolet light), $10^{17}, \ldots, 10^{20}$ Hz (X-ray emission), and $10^{20}, \ldots, 10^{22}$ Hz ($\gamma$-radiation).

Many researchers have proved [74–77] that the energy processes in plasma are of broadband nature with the frequency spectrum of $10^4, \ldots, 10^{20}$ Hz, as a result of which it is almost impossible to cover it with one sensitive element. To meet this challenge, it is proposed to use a set of sensitive elements with some overlap of the probable frequency range of the process. Each of the elements may be considered as a linear filter tuned to the specific frequency subrange. In mathematical terms, this can be described as follows:

$$\Omega_n \cong \bigcup_{i=1}^{M} \Omega_i \qquad (1)$$

where $\Omega_n$ is the frequency spectrum of the process; $\Omega_i$ is a potential fragment of the process spectrum contained in the signal at the output of the $i$th sensitive element (of the coil).

$$y(t) = \sum_{r=1}^{\infty} a_r e^{\delta_r t} + \sum_{k=1}^{\infty} \left[ A_k e^{(\delta_k + i2\pi f_k)t} + A_k^* e^{(\delta_k + i2\pi f_k)t} \right] \qquad (2)$$

where $a_r$, is the amplitude; $f_k$ is the frequency; $t$ is the time (sec); $\delta_k$ is the attenuation factor.

A computational experiment was carried out in order to check the efficiency of the spectral analysis process. A model signal, which is the sum of cosine waves and noise, was used as a test signal:

$$y(t) = \sum_{k=1}^{8} A_k \cos(2\pi f_k) + \varepsilon(t) \qquad (3)$$

The values of natural frequencies and their corresponding amplitudes, which are the parameters of the model, are contained in Table 1.

**Table 1.** Values of natural frequencies and their corresponding amplitudes, which are the parameters of the model.

| $A_k$ | 0.7 | 1.0 | 1.5 | 0.5 | 0.6 | 1.0 | 1.0 | 1.0 |
|---|---|---|---|---|---|---|---|---|
| $f_k$, Hz | 10,000 | 12,000 | 15,000 | 20,000 | 25,000 | 50,000 | 75,000 | 100,000 |

To carry out experimental studies, a broadband spectrum analyzer of electromagnetic radiation from glow discharge plasma with time-frequency signal separation was developed. The spectrum analyzer (Figure 3) consists of a multichannel measuring unit, an analog-to-digital converter (ADC), and a software module, carrying out the Prony–Fourier (PF) spectral analysis [78–84].

The measuring unit is made in the form of a set of 18 inductive sensors, which are coils of copper wire, evenly spaced along the periphery of a disk made of dielectric material. During the investigation, the measuring unit was installed at the viewing window of the vacuum chamber of the PVD unit [85–87]. To reduce the influence of external noise on the sensor, a screen in the form of a Faraday cup with grounding is installed on the unit body [84–86].

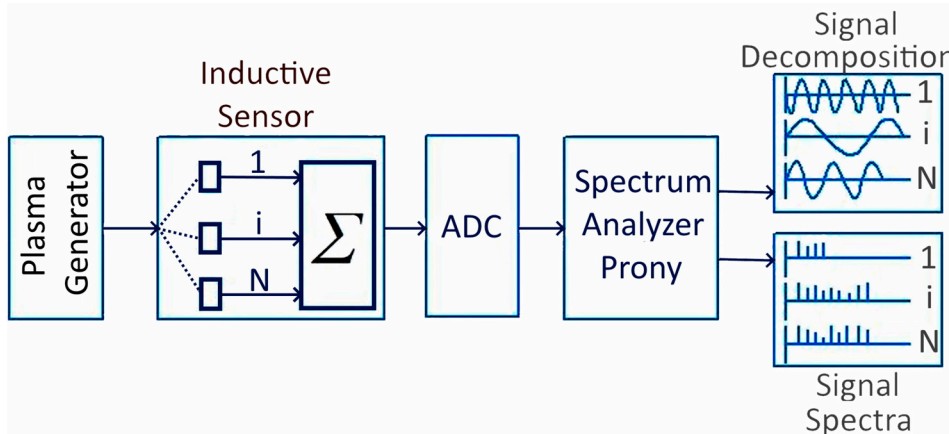

**Figure 3.** Structural diagram of the broadband Prony spectrum analyzer of electromagnetic radiation from glow discharge plasma.

The measuring unit is a distributed system with multiple inputs which number is equal to the number of coils and one output. To identify the transfer function from each input to the output, a special procedure was developed, embedded in the software module. Each coil of the measuring unit possesses a set of natural frequencies and perceives (resonates) a certain spectrum of electromagnetic radiation of the glow discharge plasma, thus acting as a physical bandpass filter. Further, the signals coming from all the sensor coils are summed up and fed through one channel to the ADC and then to the Prony spectrum analyzer, which performs the frequency-time separation of the signal. At the output of the Prony spectrum analyzer, signal frequency estimates, an analytical Fourier spectrum, an analytical energy spectrum of a signal, and its analytical time decomposition over specified frequency ranges can be obtained.

## 3. Results and Discussion

### 3.1. Analysis of the Energy Distribution of Ions in Two-Component Plasma Using a Multigrid Probe

The retarding characteristics of ions were obtained at the pressure in the vacuum chamber of ~$6 \times 10^{-3}$ Pa and titanium cathode currents reaching 180 A and 300 A. The magnitude of the magnetic field near the cathode surface did not exceed $8 \times 10^{-4}$ T. The probe was installed at a distance of 150 mm from the cathode cutoff. Figure 4 exhibits the energy distribution of titanium ions.

At arc currents of 150, . . . , 180 A, the average energy of titanium ions was ~35, . . . , 40 eV, and at current of 300 A, the ion energy decreased to ~30 eV. The average ion energy decreases, as with an increase in the arc discharge current, the number of high-energy ions in the plasma flow decreases, and the energy of titanium ions shifts to lower values. This occurs due to an increase in the evaporation rate and plasma concentration and, consequently, the degree of plasma flow randomness in the near-cathode region due to an increase in the number of particle collisions.

In case of sublimating materials with high saturated vapor pressure (C, Cr), an evaporable anode was made in the form of a rod. An anode in the form of a crucible made of refractory materials was used in case of substances with low vapor pressure. In the describe case, the anode was a rod 6 mm in diameter made of spectrally pure carbon. To isolate graphite plasma ions from the two-component plasma, a screen was installed in front of the probe diaphragm, at a distance of 50 mm from the cathode surface, under a floating potential, which prevented titanium ions generated by the cathode spot from entering the analyzer. The probe characteristics were taken at two values of the electron current to the evaporable anode: 70 A and 120 A. In both cases, the cathode current was 150 A. The burning voltage of the arc discharge with the evaporable graphite anode was 40–43 V when

the anode current varied from 50 to 150 A, respectively. The energy distribution curves of carbon ions generated by the evaporable anode are exhibited in Figure 3.

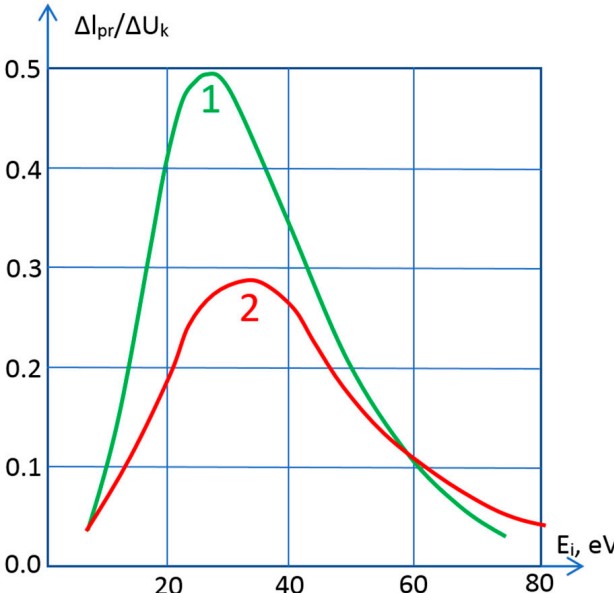

**Figure 4.** Energy distribution of titanium ions. 1—discharge current of 180 A, 2—discharge current of 300 A.

The energy spectrum of the ions of the cathode material almost coincides with the spectrum of ions during the operation of the arc evaporator in the normal mode without a sacrificial anode.

The estimative analysis of the spectrum of carbon ions generated by the anode (Figure 5) finds that with an increase in the anode current, the average energy of carbon ions decreases noticeably. Therefore, when the anode current is 70 A, the average ion energy reaches ~15, . . . , 18 eV, and as the current increases to 120 A, it decreases to ~9, . . . , 10 eV.

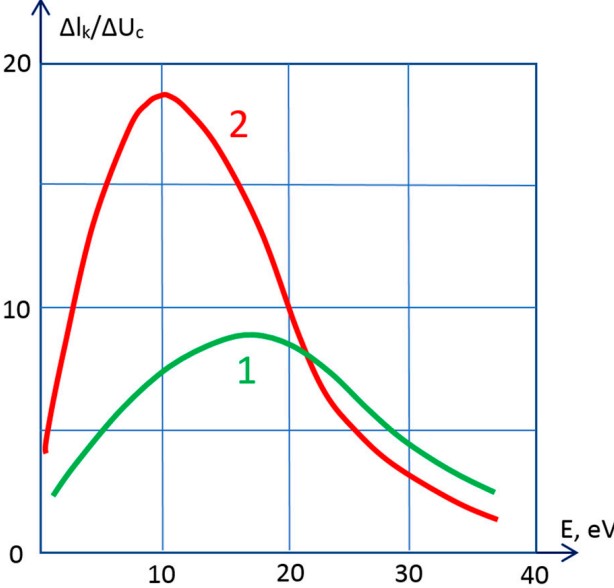

**Figure 5.** Energy distribution of carbon ions. 1—anode current of 70 A, 2—anode current of 120 A.

The effect of the decrease in the energy of graphite ions may occur, as with an increase in the anode current, the temperature of the anode surface, the rate of evaporation of graphite, and, consequently, the density of plasma and vapor increase. The growth of the plasma density and, importantly, vapor density leads to a decrease in the degree of plasma ionization and an increase in the randomness of particle motion in the plasma flow due to an increase in the number of collisions of ions with particles having thermal velocities. This is confirmed by the fact that at low anode currents, not exceeding 70 A, the average energy of graphite ions correlates well with the anode potential.

### 3.2. Plotting Curves of the Distribution of Argon Ions Using a Source of Gas Ions with Closed Electron Current

Gas ion sources with a closed electron current are an effective tool for the implementation of ionic surface treatment processes [88–92]. Technological ion sources are used for cleaning and activating the surface of products before applying thin-film coatings, ion assistance in the processes of coating deposition, ion-beam etching, and shaping.

There is a high-voltage mode of operation of ion sources with the discharge voltage from 600 to 3000 V and higher and a low-voltage mode with the voltage usually up to 300–500 V. The first mode is usually applied in the processes of depositing thin layers by target sputtering and for cleaning and etching surfaces. The second mode is used in the processes of coating deposition with etching and in the processes of ion cleaning. The energy of accelerated ions generated by a source is determined by the operating voltage of the discharge. Figure 6 depicts the energy distribution curves of argon ions depending on the discharge voltage for the low-voltage mode of operation of the ion source, obtained using a multigrid probe [93–95].

It can be noted from the curves that the average ion energy decreases with a decline in the discharge current, and the source efficiency (the ratio of the average ion energy W to the discharge voltage U) remains approximately at the same level in the operating voltage range of the source ions.

For carrying out the processes of coating deposition with etching, an energy value from 100 to 200 eV is sufficient. A built-in multigrid probe analyzer of ion energy may be used to adjust the technological processes of ion processing. For some ion treatment processes (for example, surface cleaning), the above ion energy is not enough. To increase the energy of the bombarding ions and improve the efficiency of etching and cleaning, a negative potential (bias potential) can be applied to the substrate.

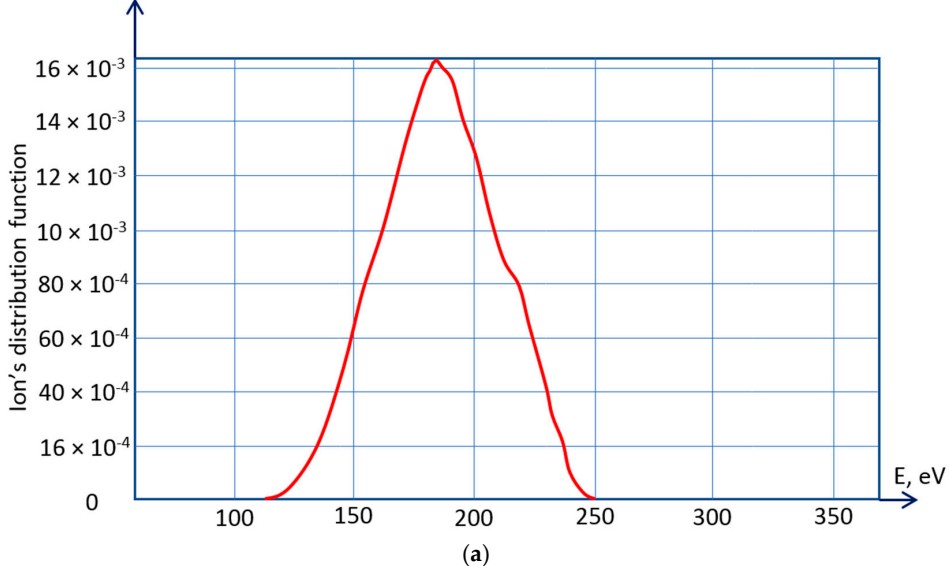

(a)

**Figure 6.** *Cont.*

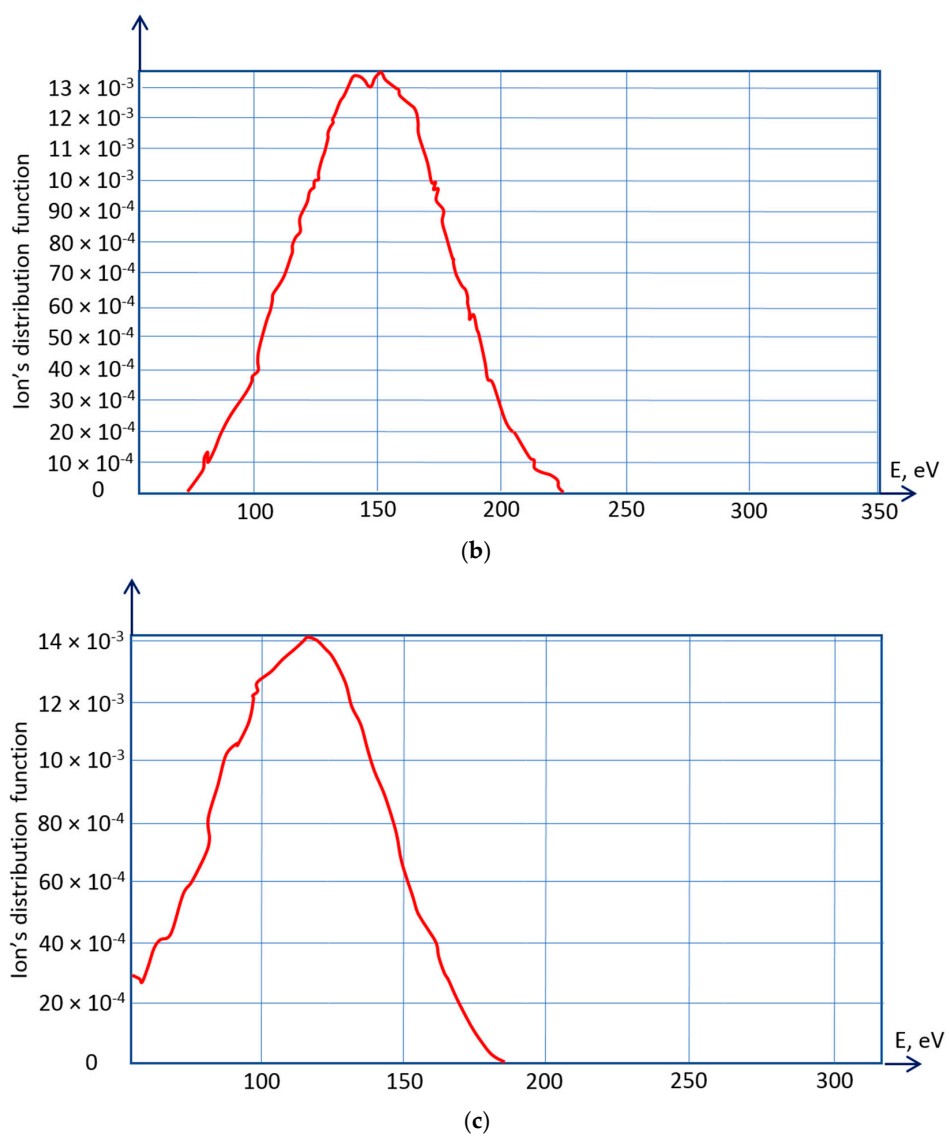

**Figure 6.** Distribution curves of argon ions depending on the energy of the low-voltage ion source. At discharge voltage of: (**a**) −250 V, (**b**) −230 V, (**c**) −200 V.

*3.3. Spectrum Analyzer of the Signal of Electromagnetic Radiation of Glow Discharge Plasma Based on Multichannel Inductive Sensor*

Figure 7 exhibits a graphical relationship between the oscillatory process synthesized in accordance with relationship (3) and Table 1.

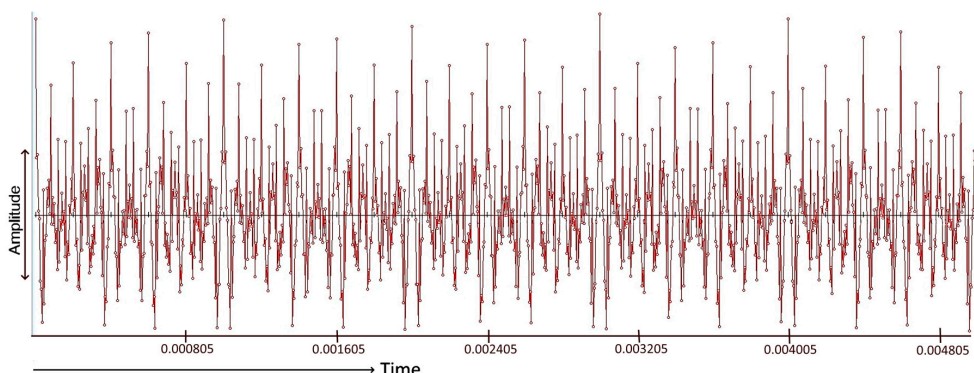

**Figure 7.** Relationship between the synthesized test signal of the oscillatory process and the time.

Since the linear bandpass filters in the model signal are tuned to the following frequency subranges: $(5, \ldots, 18)$, $(18, \ldots, 40)$, $(40, \ldots, 120)$ kHz, then its analytical decomposition is tuned to the corresponding subranges. The results of the analytical modal decomposition are presented in Figure 8. The left side of the figure shows graphs displaying all modes from the corresponding specified frequency, and the right side of the spectrum is $A^2$, squared Prony amplitudes [85,87].

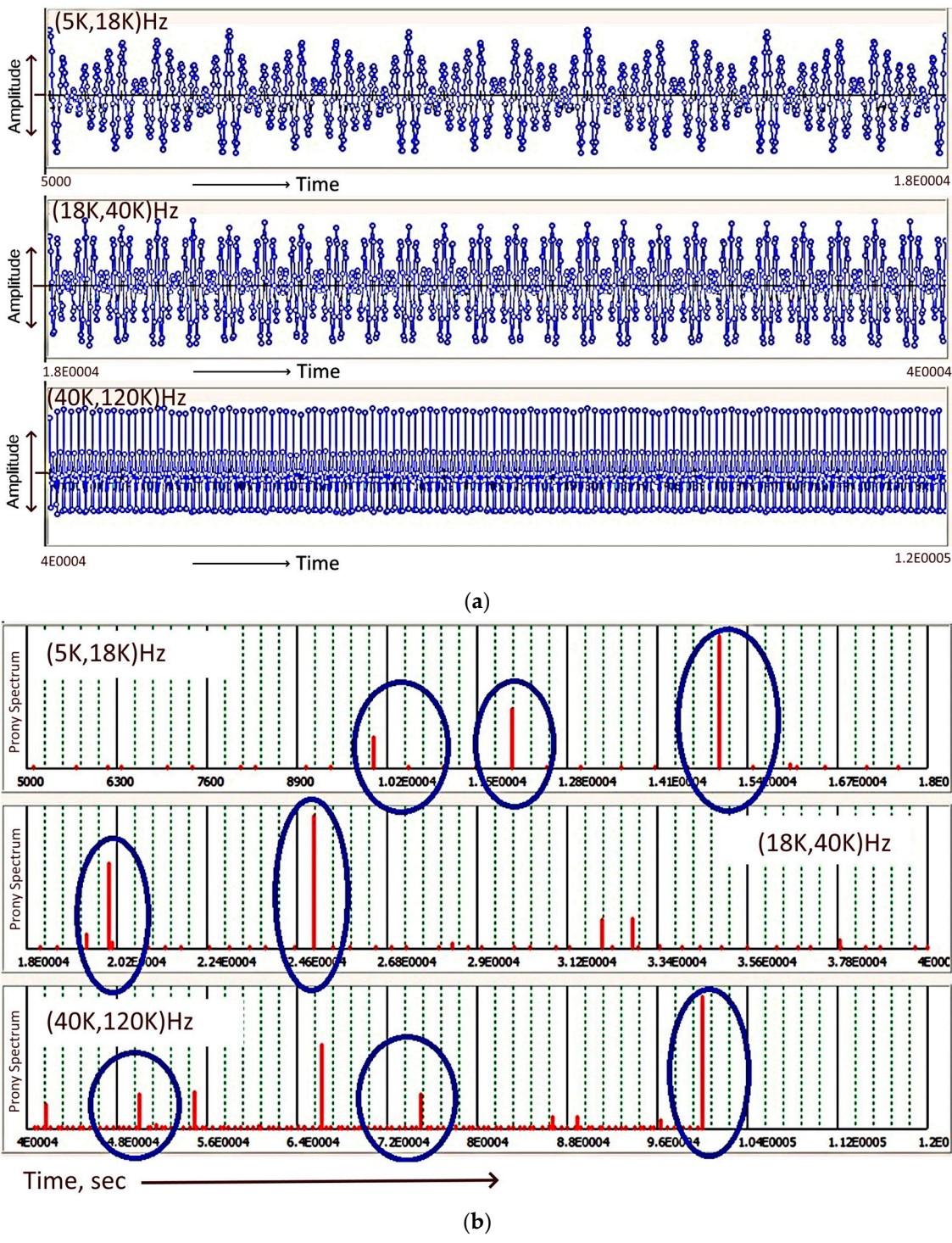

**Figure 8.** Decomposition of the signal by the frequency subbands of the sensors. (**a**) time signal, (**b**) its $A^2$—the Prony spectrum. Model signal modes are circled with blue ovals [79,80].

Modes of the model signal are selected on the $A^2$ Prony spectrum (circled with blue ovals), and the remaining methods correspond to the random component or are mathematical models, that is, they are not actually present in the signal, but appear as a result of signal approximation.

Figure 9 shows a stabilization diagram for the frequency estimates of the model signal. The stabilization diagram in this case of estimates of natural frequencies shows the stability of the appearance of a mode with the corresponding natural frequency for a given order (maximum number of modes) of the approximation model (1). Physical modes i.e., actually present in the signal are characterized by vertical lines already after the number of modes equal to 15. The stabilization diagram clearly showed modes with model frequencies, namely: 10, 12, 15, 20, 25, 50, 75, and 100 kHz.

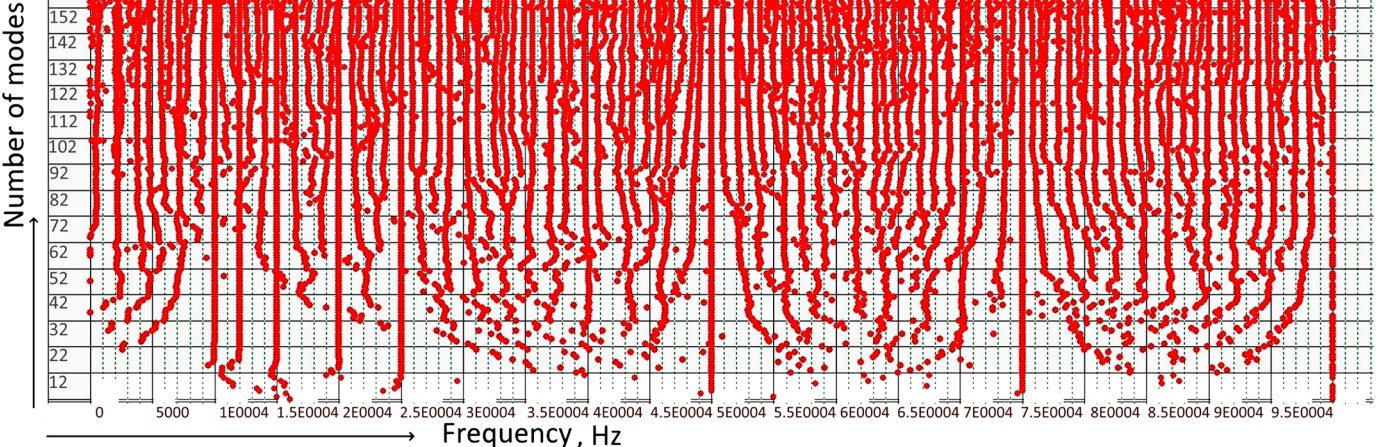

**Figure 9.** Stabilization diagram for frequency estimates of the model process. The horizontal axis exhibits frequency, and the vertical axis—the number of modes (model complexity), i.e., the number of components in the approximation model, which may be much larger than in the original test model.

Figure 10 shows the analytical Fourier spectrum of the total model time signal, the spectral lines of which correspond with high accuracy to the original natural frequencies of the model. The analytical Fourier spectrum (Prony–Fourier spectrum) is the result of the analytical integration of the signal model (1) obtained as a result of the Prony approximation procedure [86,87]. The analytical Prony–Fourier spectrum has an increased resolution compared to conventional discrete Fourier transforms.

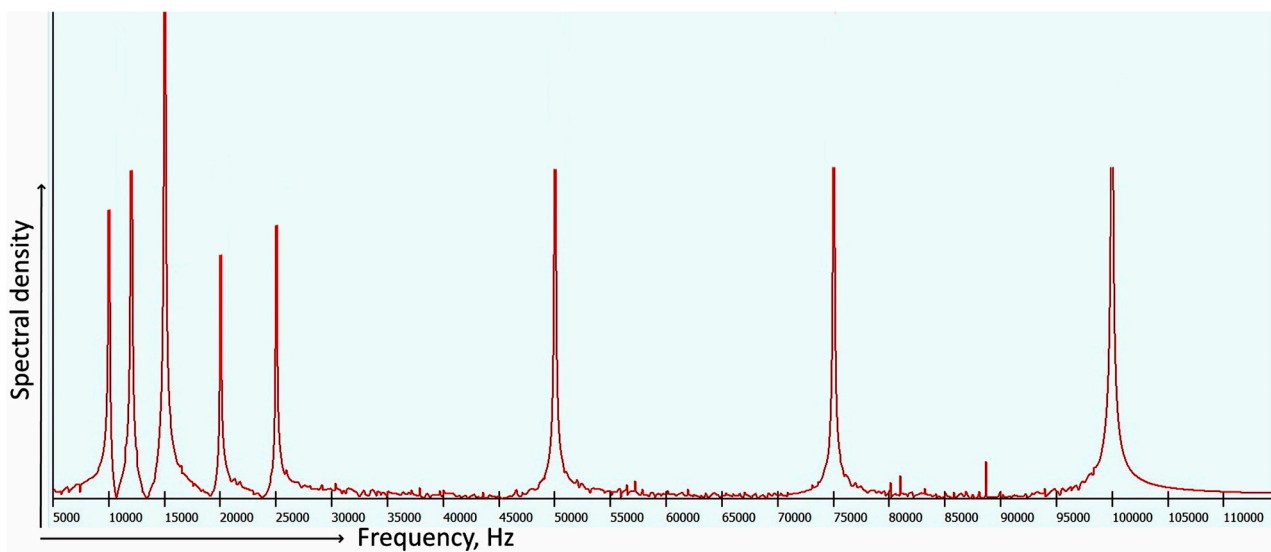

**Figure 10.** Spectrum of the total model signal.

Figure 11 shows a graphical dependence of the recorded changes in the electromagnetic radiation of the glow discharge plasma, obtained at the output of the ADC during the processing in the PVD installation, and in Figure 12a,b, respectively, the analytical Fourier spectrum (spectral density) and the analytical energy spectrum of the signal after processing in the Prony spectrum analyzer.

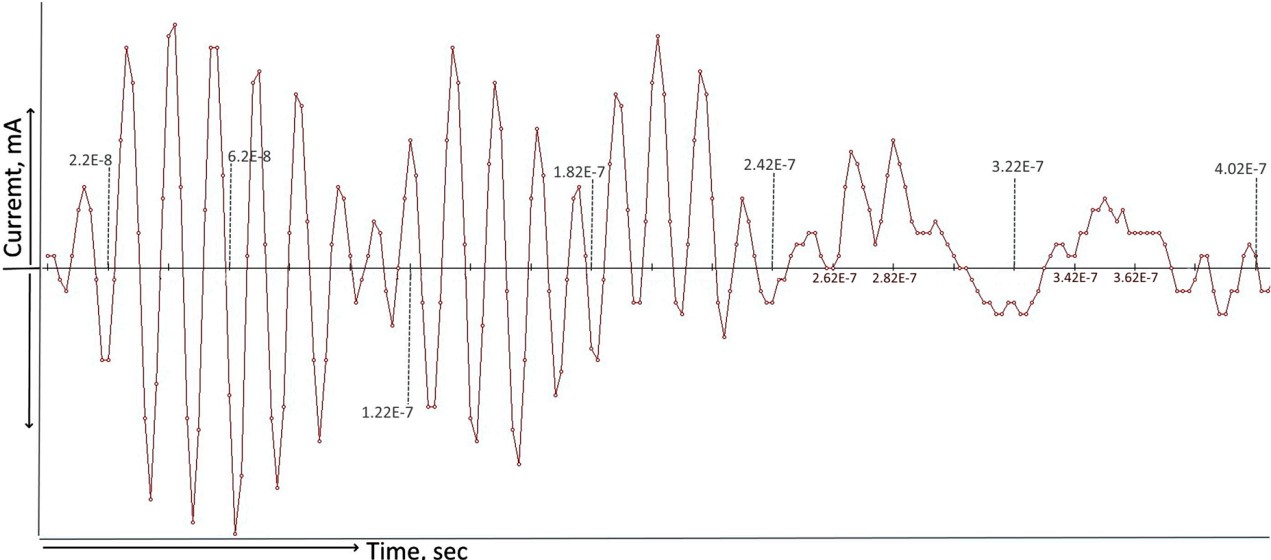

**Figure 11.** Graphical relationship between the changes in the electromagnetic radiation from the glow discharge plasma obtained at the output of the ADC.

An analysis of the spectra presented in Figure 12 shows that their structure is saturated and contains several clearly distinguishable frequencies, i.e., at a qualitative level, it is quite consistent with the ongoing physical processes. For final conclusions about the effectiveness of the proposed scheme of the spectrum analyzer, additional experimental studies are planned.

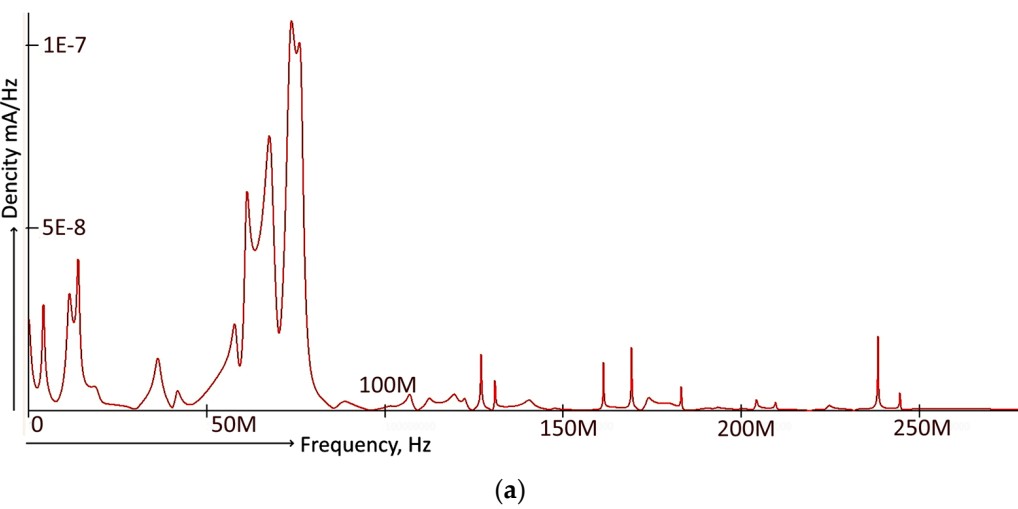

(a)

**Figure 12.** *Cont.*

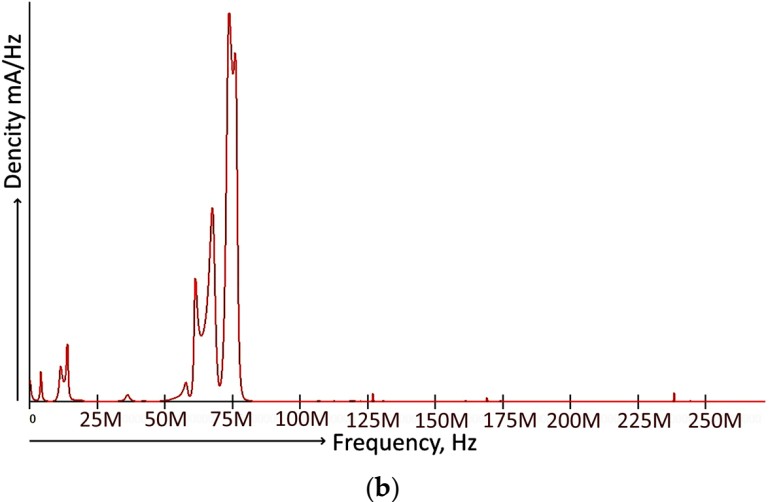

(**b**)

**Figure 12.** (**a**) Analytical Fourier spectrum of the signal at the output of the Prony spectrum analyzer (relationship between the amplitude spectrum of the signal (density mA/Hz) and frequency, Hz) and (**b**) its energy spectrum.

### 4. Conclusions

The article describes the main advantages and disadvantages of the methods in relation to the continuity of vacuum technological processes directed to modification of surfaces of solid bodies.

1.  The advantage of the probe methods for plasma diagnostics is the information content and ease of their adaptation to technological equipment.
2.  The possibility of determining the energy distribution of ions in the two-component plasma has been described. The energy distribution curves of ions in the two-component plasma generated by a vacuum arc evaporator with sacrificial cathode and anode have been plotted.
3.  Energy distribution curves of argon ions in the low-voltage mode of operation of ion sources with a closed electron current have been analyzed. The average ion energy decreases with a decline in the discharge current, and the source efficiency (the ratio of the average ion energy W to the discharge voltage U) remains approximately at the same level of W/U ≈ 0.68, . . . , 0.71 in the operating voltage range of the source.
4.  A scheme of a spectrum analyzer is proposed, which can be used both for monitoring and for controlling the processing process, including in automated PVD installations.
5.  The proposed magnetic induction method for monitoring the glow discharge plasma parameters can be used to determine the moment the PVD facility enters the operating mode and completes the process steps, as well as to perform operational intervention in the process, taking into account changes in electromagnetic pulses.
6.  The proposed electromagnetic radiation signal spectrum analyzer based on a multichannel inductive sensor makes it possible to analyze broadband plasma electromagnetic radiation signals with the possibility of obtaining not only spectra at the sensor output, but also deconvolution of the glow discharge plasma electromagnetic radiation signal spectrum.

Thus, it can be argued that the proposed scheme of the spectrum analyzer is fundamentally operable, however, its verification requires specially planned experiments.

**Author Contributions:** Conceptualization, S.D. and A.V.; methodology, V.Z.; investigation, S.D., C.S. and V.Z.; resources, S.G.; data curation, A.V.; writing—original draft preparation, A.V., S.D. and V.Z.; writing—review and editing, A.V.; project administration, S.G.; funding acquisition, S.G. All authors have read and agreed to the published version of the manuscript.

**Funding:** This work was supported financially by the Ministry of Science and Higher Education of the Russian Federation (project No FSFS-2021-0003).

**Institutional Review Board Statement:** Not applicable.

**Informed Consent Statement:** Not applicable.

**Data Availability Statement:** Not applicable.

**Conflicts of Interest:** The authors declare no conflict of interest.

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
