# Peer review of "Diagnostic Techniques for Electrical Discharge Plasma Used in PVD Coating Processes"

_coatings, doi:10.3390/coatings13010147_

Round 1
Reviewer 1 Report
The authors introduced the diagnostic techniques for electrical discharge plasma used in PVD coating processes. They first reviewed the methods for analyzing plasma parameters in the PVD technologies designed for surface modification and deposition of thin films and coatings, then discussed the two-component plasma and the broadband signals of plasma electromagnetic radiation. It is a bit confusing that the half part is review, though generally people review the past works to point out the challenges and issues in the introduction part, then explain why the research is important. Here the content makes readers feeling like the manuscript is formed by two parts: a short review and some experimental data analysis. The language needs to be significantly improved, the organization of content and the deeper analysis of the results are necessary, thus I may suggest rejection.
1. The text should be well refined and be more concise. E.g., for introduction part and part 2, the roadmap from previous/current research to the solution provided in this article should be clearer.
2. For figures, the authors are suggested unifying the font, making the schematic clearer, such as Figure 2, providing unit in figures, and clearer description in caption. The quality of the figures is quite poor (I would suggest replotting the figures with original data, making the figures more meaningful and emphasizing the key points), and some of them are not essential to be placed into different figures. Figure 6-12 are just presented, while further analysis is missing.
Author Response
Reviewer 1
R: The authors introduced the diagnostic techniques for electrical discharge plasma used in PVD coating processes. They first reviewed the methods for analyzing plasma parameters in the PVD technologies designed for surface modification and deposition of thin films and coatings, then discussed the two-component plasma and the broadband signals of plasma electromagnetic radiation.
A: The authors are grateful to the Reviewer for his help in improving the quality of the manuscript.
R: It is a bit confusing that the half part is review, though generally people review the past works to point out the challenges and issues in the introduction part, then explain why the research is important. Here the content makes readers feeling like the manuscript is formed by two parts: a short review and some experimental data analysis.
A: Since the article deals with two different methods (Prony analyzer and Langmuir probe measurements), the authors found it useful to give some overview of the different methods for monitoring the state of the plasma. The authors revised the Introduction, reducing redundant information and expanding the description of the features of the methods considered in the article.
R: The language needs to be significantly improved,
A: The English language has been further checked and a number of corrections have been made.
R: the organization of content and the deeper analysis of the results are necessary, thus I may suggest rejection.
A: Additional analysis of the information provided was added in accordance with the recommendations of the Reviewer.
- The text should be well refined and be more concise. E.g., for introduction part and part 2, the roadmap from previous/current research to the solution provided in this article should be clearer.
A: The Introduction and Materials and Methods sections have been substantially modified.
R: 2. For figures, the authors are suggested unifying the font, making the schematic clearer, such as Figure 2, providing unit in figures, and clearer description in caption. The quality of the figures is quite poor (I would suggest replotting the figures with original data, making the figures more meaningful and emphasizing the key points), and some of them are not essential to be placed into different figures.
A: The figures were redrawn, the resolution of the figures was increased, the font was unified, the key inscriptions were enlarged.
R: Figure 6-12 are just presented, while further analysis is missing.
A: A more detailed analysis of these figures has been added.
Reviewer 2 Report
Dear Authors,
The manuscript coatings-2076821, with the title 'Diagnostic techniques for electrical discharge plasma used in PVD coating processes' presents several techniques for analyzing plasma parameters during deposition. The possibility of determining the energy distribution of the ions is discussed and the corresponding curves are presented, in an arc vacuum evaporation installation. The results are focused on the analysis of the electromagnetic signals radiated by the plasma, using inductive multichannel analyzers. The text of the manuscript seems relatively well written, with a good positioning and structuring of the figures of the presented methods. However, the quality of the images is not the best, and should be much improved for a final acceptance phase. The structuring of the text, as a whole, is not a very happy one, being unclear the transition from the theoretical presentation of the methods to the part of experimental examples and the part of results combined with discussions. A concrete division of the text into paragraphs containing information related to: methods, results, discussions, must be done, in order to easily and clearly follow these sections of the work.
It is uncertain at this level, and in this form of the manuscript, which of the figures represent experimental data and which only theoretical concepts.
In its current form, the manuscript cannot be considered for publication in Coatings.
I recommend REJECT with Resubmission, following the agreed structure of the journal (author guidelines), respectively following the suggestions given above.
Author Response
Reviewer 1
R: The manuscript coatings-2076821, with the title 'Diagnostic techniques for electrical discharge plasma used in PVD coating processes' presents several techniques for analyzing plasma parameters during deposition. The possibility of determining the energy distribution of the ions is discussed and the corresponding curves are presented, in an arc vacuum evaporation installation. The results are focused on the analysis of the electromagnetic signals radiated by the plasma, using inductive multichannel analyzers. The text of the manuscript seems relatively well written, with a good positioning and structuring of the figures of the presented methods.
A: The authors are grateful to the Reviewer for the high appreciation of their work. The authors tried to take into account all the recommendations of the Reviewer when preparing an updated version of the manuscript.
R: However, the quality of the images is not the best, and should be much improved for a final acceptance phase.
A: The authors apologize for the deterioration in the quality of the images when building the pdf version of the article. The resolution of the images has been increased and the original images in JPEG format are attached to the manuscript.
R: The structuring of the text, as a whole, is not a very happy one, being unclear the transition from the theoretical presentation of the methods to the part of experimental examples and the part of results combined with discussions. A concrete division of the text into paragraphs containing information related to: methods, results, discussions, must be done, in order to easily and clearly follow these sections of the work.
A: The authors revised the structure of the manuscript in accordance with the recommendations of the Reviewer, with which they fully agree. Now the structure of the manuscript is as follows:
- Introduction
1.1. Method of physical vapor deposition and issues of process diagnostics.
1.2. Key methods of operational control of the deposition process parameters
- Materials and methods
2.1. The Langmuir probe measurements
2.2. Broadband spectrum analyzer
- Results and discussion.
3.1. Analysis of the energy distribution of ions in two-component plasma using a multigrid probe
3.2. Plotting curves of the distribution of argon ions using a source of gas ions with closed electron current
3.3. Spectrum analyzer of the signal of electromagnetic radiation of glow discharge plasma based on multichannel inductive sensor
- Conclusions
- It is uncertain at this level, and in this form of the manuscript, which of the figures represent experimental data and which only theoretical concepts.
- After a corresponding change in the structure of the manuscript, it is logically determined that the figures in section 2 describe the scheme of the devices used, and the figures in section 3 represent the results obtained as a result of the research.
The authors hope that after the changes made, the manuscript meets the requirements for publication. The authors once again thank the Reviewer for useful recommendations.
Reviewer 3 Report
The article talks about several techniques of electrical discharge plasma diagnostics used 13
in the Physical vapor deposition (PVD) coating processes. Different methods for analyzing plasma parameters in the PVD technologies designed for surface modification and deposition of thin films and coatings has been reviewed.. The paper has some interesting results that could make it publishable in the Journal of Coatings after the following major revisions:
1-The language of the paper needs modification. Please check the English language of the paper.
2-The last sentence of the abstract should be modified. It is now very long and tedious.
3-Define in the abstract what parameters were investigated, what kind of tests were employed and briefly mention the results of such tests. You mentioned part of it, but needs to be a complete one.
4-Introduciton should be strengthened. To modify this section the following documents can be consulted:
- Theoretical study of novel B–C–O compounds with non-diamond isoelectronic. Chinese Physics B, 31(2), 26201. doi: 10.1088/1674-1056/ac0cd2
- An eco-friendly film of pH-responsive indicators for smart packaging. Journal of Food Engineering, 321, 110943. doi: https://doi.org/10.1016/j.jfoodeng.2022.110943
-(2021). Ultra-fast growth of cuprate superconducting films: Dual-phase liquid assisted epitaxy and strong flux pinning. Materials Today Physics, 18, 100400. doi: 10.1016/j.mtphys.2021.100400
5-Figure 1 it should be schematic not scheme.
6-Perhaps a new section to be created regarding the type of analysis and the methods of analyzing.
8-consult the following reference in the discussion section
-(2022). An asymmetric encoder–decoder model for Zn-ion battery lifetime prediction. Energy Reports, 8, 33-50. doi: https://doi.org/10.1016/j.egyr.2022.09.211
9-figures 11, and 12 are of low quality.
10-Conlsuions are long. A more brief conclusions should be presented.
Author Response
Reviewer 2
The article talks about several techniques of electrical discharge plasma diagnostics used in the Physical vapor deposition (PVD) coating processes. Different methods for analyzing plasma parameters in the PVD technologies designed for surface modification and deposition of thin films and coatings has been reviewed.. The paper has some interesting results that could make it publishable in the Journal of Coatings after the following major revisions:
A: The authors are grateful to the Reviewer for the high appreciation of their work. The authors tried to take into account all the recommendations of the Reviewer when preparing an updated version of the manuscript.
R1-The language of the paper needs modification. Please check the English language of the paper.
A: The English language was additionally corrected and the necessary changes made.
R2-The last sentence of the abstract should be modified. It is now very long and tedious.
A: The phrase is divided into two parts and now, as it seems to the authors, it has become easier to understand.
R: 3-Define in the abstract what parameters were investigated, what kind of tests were employed and briefly mention the results of such tests. You mentioned part of it, but needs to be a complete one.
A: The abstract has been added
R4-Introduciton should be strengthened. To modify this section the following documents can be consulted:
- Theoretical study of novel B–C–O compounds with non-diamond isoelectronic. Chinese Physics B, 31(2), 26201. doi: 10.1088/1674-1056/ac0cd2
- An eco-friendly film of pH-responsive indicators for smart packaging. Journal of Food Engineering, 321, 110943. doi: https://doi.org/10.1016/j.jfoodeng.2022.110943
-(2021). Ultra-fast growth of cuprate superconducting films: Dual-phase liquid assisted epitaxy and strong flux pinning. Materials Today Physics, 18, 100400. doi: 10.1016/j.mtphys.2021.100400
A: The proposed works are taken into account in the Introduction
R: 5-Figure 1 it should be schematic not scheme.
A: The authors agree with the Reviewer. Changes applied.
R: 6-Perhaps a new section to be created regarding the type of analysis and the methods of analyzing.
A: Section Materials and Methods added.
R: 8-consult the following reference in the discussion section
-(2022). An asymmetric encoder–decoder model for Zn-ion battery lifetime prediction. Energy Reports, 8, 33-50. doi: https://doi.org/10.1016/j.egyr.2022.09.211
A: The work proposed by the Reviewer was taken into account.
R: 9-figures 11, and 12 are of low quality.
A: The authors apologize for the deterioration in the quality of the images when building the pdf version of the article. The resolution of the images has been increased and the original images in JPEG format are attached to the manuscript.
R: 10-Conlsuions are long. A more brief conclusions should be presented.
A: The conclusion has been shortened.
Round 2
Reviewer 1 Report
The authors have revised the manuscript and the quality is improved compared with the previous versions. It is appreciable that the effort has been made to provide a clearer and well-organized draft, while this work is still not qualified for publication yet. I also noticed that the Reviewer 2 is quite patient and has provided plenty of concrete guidance on how to form a sound paper. I do suggest that the authors strictly follow the comments thus reducing the number of review rounds. I would agree the publication if the authors fully address the comments as following and Reviewer 2’s comments in Round 2.
1. For abstract, e.g., the first sentence is strange “The article deals with several techniques of electrical discharge plasma diagnostics used in the Physical vapor deposition (PVD) coating processes”, which is not a good starting point and a bit misleading. In Page 4, it describes that the paper mainly discusses two methods. The second sentence also does not well fit with the context. The authors are suggested to directly and concisely provide “why” you start this research with these two methods for the “review”, the significant benefits, rather than just saying what has been done. The following part can be shortened a bit.
2. The structure is clearer while the authors should pay more attention to the language, e.g., Page 3 Line 131 “your own frequency spectrum of the segment of the time series” is colloquial. In addition, it’s not necessary to have so many paragraphs. E.g., Line 133-136.
3. The figures are still not fine enough. E.g., Figure 2, the same color, and similar pattern used for the different parts of probe are misleading. Figure 3, “Inductive Sensor” is stretched horizontally, unify font type font size, capitalization. What does “relative units” mean in Figure 7? The numbers on x axis label are meaningless as the readers cannot distinguish them, also in Figures 8-12. It gives the feeling that the figures are screenshot rather than plotted with data. The authors may refer to references like 21, 69, 71, 92 to clearer present the data and figures.
Author Response
R: The authors have revised the manuscript and the quality is improved compared with the previous versions. It is appreciable that the effort has been made to provide a clearer and well-organized draft, while this work is still not qualified for publication yet. I also noticed that the Reviewer 2 is quite patient and has provided plenty of concrete guidance on how to form a sound paper. I do suggest that the authors strictly follow the comments thus reducing the number of review rounds. I would agree the publication if the authors fully address the comments as following and Reviewer 2’s comments in Round 2.
A: The authors are grateful to the Reviewer for their invaluable help and careful analysis, which helps to improve the quality of the manuscript. The authors tried to take into account all the comments of the Reviewer when preparing the corrected version of the manuscript, and also re-examined the recommendations of Reviewer 2 and made the appropriate corrections in the manuscript.
R: 1. For abstract, e.g., the first sentence is strange “The article deals with several techniques of electrical discharge plasma diagnostics used in the Physical vapor deposition (PVD) coating processes”, which is not a good starting point and a bit misleading. In Page 4, it describes that the paper mainly discusses two methods. The second sentence also does not well fit with the context. The authors are suggested to directly and concisely provide “why” you start this research with these two methods for the “review”, the significant benefits, rather than just saying what has been done. The following part can be shortened a bit.
A: The abstract has been substantially modified in accordance with the recommendations of the Reviewer.
R: 2. The structure is clearer while the authors should pay more attention to the language, e.g., Page 3 Line 131 “your own frequency spectrum of the segment of the time series” is colloquial.
A: The phrase has been revised
R: In addition, it’s not necessary to have so many paragraphs. E.g., Line 133-136.
A: The number of paragraphs has been revised in this and other fragments of the manuscript.
R: 3. The figures are still not fine enough. E.g., Figure 2, the same color, and similar pattern used for the different parts of probe are misleading.
A: Fig 2 has been modified in accordance with the recommendations of the Reviewer
R: Figure 3, “Inductive Sensor” is stretched horizontally, unify font type font size, capitalization.
A: Figure 3 has been modified in accordance with the recommendations of the Reviewer
R: What does “relative units” mean in Figure 7?
A: This refers to the relative spread of the syntesizxed test signal of the oscillatory process. The designation "relative units" has been removed, since it has no physical meaning.
R: The numbers on x axis label are meaningless as the readers cannot distinguish them, also in Figures 8-12. It gives the feeling that the figures are screenshot rather than plotted with data. The authors may refer to references like 21, 69, 71, 92 to clearer present the data and figures
A: Figure 7 - 12 has been further modified - small unreadable characters have been removed and replaced with larger ones where possible.
Reviewer 2 Report
Dear Authors,
The revised manuscript is improved compared to the first version, however, in its present form it cannot be considered for publication in the Coatings journal.
Concretely, even if the manuscript was divided into sections, there is still a lot to be done with the text and images in order to have a manuscript to consider. In the figures, the text of the sizes on the axes must be enlarged to almost the text of the manuscript (so that they can be visualized and easily read by the reader), specifically figures 4 to 12. Figure 8 (left and right graphs) must be enlarged and explained much better the insets made with the blue ellipses, with a better correlation with lines 359-361 of the text.
To be explained, because it is still not clear, what this diagnostic method brings as novelty and information compared to other diagnostic techniques on this plasma. What would be the advantages/disadvantages of this method compared to the classic methods of diagnosis. A parallel between this method and the conventional ones.
The conclusions are extraordinarily narrow, even if 6 broad lines are listed in this section - which should be integrated in the discussion part.
In its current revised form, it should not yet be considered for publication.
The manuscript should include an introduction to the field, at the end of which the "problem and the method by which the problem is solved" should be outlined by the authors. There should be a clear section of materials and method/methods. There should be a section dedicated to the experimentally obtained / modeled results (with a clear expression of what is experimental and what is obtained by modeling). There should be a discussion section, if not included in 'Results and discussions'. At the end, a section of conclusions, with a transient conclusion summarizing the results, followed, optionally, by the most important results obtained, and the perspective.
Rejected with resubmission.
Author Response
Reviewer 2
R: The revised manuscript is improved compared to the first version, however, in its present form it cannot be considered for publication in the Coatings journal.
A: The authors are grateful to the Reviewer for his help in improving the quality of the manuscript.
R: Concretely, even if the manuscript was divided into sections, there is still a lot to be done with the text and images in order to have a manuscript to consider. In the figures, the text of the sizes on the axes must be enlarged to almost the text of the manuscript (so that they can be visualized and easily read by the reader), specifically figures 4 to 12.
A: The figures have been redrawn, their resolution has been increased, the fonts have been unified, the size of the key text data has been increased for ease of understanding.
R: Figure 8 (left and right graphs) must be enlarged and explained much better the insets made with the blue ellipses, with a better correlation with lines 359-361 of the text.
A: Figure 8 has been enlarged, the required description has been expanded.
R: To be explained, because it is still not clear, what this diagnostic method brings as novelty and information compared to other diagnostic techniques on this plasma. What would be the advantages/disadvantages of this method compared to the classic methods of diagnosis. A parallel between this method and the conventional ones.
A: A corresponding section has been added to the Introduction, which discusses the features of this method.
R: The conclusions are extraordinarily narrow, even if 6 broad lines are listed in this section - which should be integrated in the discussion part.
A: The Conclusions section has been further modified.
R: In its current revised form, it should not yet be considered for publication. The manuscript should include an introduction to the field, at the end of which the "problem and the method by which the problem is solved" should be outlined by the authors. There should be a clear section of materials and method/methods. There should be a section dedicated to the experimentally obtained / modeled results (with a clear expression of what is experimental and what is obtained by modeling). There should be a discussion section, if not included in 'Results and discussions'. At the end, a section of conclusions, with a transient conclusion summarizing the results, followed, optionally, by the most important results obtained, and the perspective.
A: The authors additionally modified the manuscript in accordance with the recommendations of the Reviewer. The structure of the manuscript was additionally changed, redundant information that was not directly related to the subject of the manuscript was removed, and information about the methods under study was added.
Most of the Figures have been modified to make them easier for readers to understand.
The description of the results obtained was expanded, the conclusions were also modified. The authors hope that the modified form of the manuscript will better meet the criteria for acceptance for publication. At the same time, the authors are ready to continue work to improve the quality of the manuscript. The authors are grateful to the Reviewer for their help in working on the manuscript.
Round 3
Reviewer 1 Report
The structure and figures are much clearer than the initial version. It is appreciable if the authors can address the following issues before the manuscript being accepted.
In Fig. 7 and 8, appropriate y labels should be added.
For Fig. 8 a, please remove the overlapped x labels and add clear x label and scale.
It is a bit confusing that in Fig. 10, Do these frequencies. E.g., 10000, 15000, 50000, etc. share the same value of spectral density? Are these results consistent with Table 1?
Author Response
Reviewer 1
Comments and Suggestions for Authors
R: The structure and figures are much clearer than the initial version. It is appreciable if the authors can address the following issues before the manuscript being accepted.
A: The authors are grateful to the Reviewer for their attentiveness and invaluable help in improving the manuscript. The authors apologize for the initial imperfection of the manuscript, the quality of which was improved with the help of the Reviewer's recommendations.
R: In Fig. 7 and 8, appropriate y labels should be added.
A: Figures 7 and 8 have been revised and supplemented in accordance with the recommendations of the Reviewer.
R: For Fig. 8 a, please remove the overlapped x labels and add clear x label and scale.
A: Figure 8 has been changed - the small print has been removed, the designations of the initial and final values ​​on the scales have been added.
R: It is a bit confusing that in Fig. 10, Do these frequencies. E.g., 10000, 15000, 50000, etc. share the same value of spectral density? Are these results consistent with Table 1?
A: The authors rechecked Figure 10. Unfortunately, mistakes were made, which have now been corrected. The authors are grateful to the Reviewer for their help.
Reviewer 2 Report
Dear Authors,
The rev4 of the manuscript is much more better then the previous versions, but still need some improvements. Please try to put also the represented size on the vertical axis on figure 7, page 11. Same thing on figure 8, page 12. Is it the 'amplitude' of the signal that is represented vs time?
MINOR REVISION
Author Response
Reviewer 2
Comments and Suggestions for Authors
Dear Authors,
R: The rev4 of the manuscript is much more better then the previous versions, but still need some improvements.
A: The authors are grateful to the Reviewer for their attentiveness and invaluable help in improving the manuscript. The authors apologize for the initial imperfection of the manuscript, the quality of which was improved with the help of the Reviewer's recommendations.
R: Please try to put also the represented size on the vertical axis on figure 7, page 11.
A: Figure 7 have been revised and supplemented in accordance with the recommendations of the Reviewer.
R: Same thing on figure 8, page 12.
A: Figure 8 have been revised and supplemented in accordance with the recommendations of the Reviewer.
R: Is it the ' Amplitude' of the signal that is represented vs time?
A: The authors are grateful to the Reviewer - namely, Amplitude.